# Research on Adaptive Closed-Loop Control of Microelectromechanical System Gyroscopes under Temperature Disturbance

**DOI:** 10.3390/mi15091102

**Published:** 2024-08-30

**Authors:** Ke Yang, Jianhua Li, Jiajie Yang, Lixin Xu

**Affiliations:** 1School of Mechatronical Engineering, Beijing Institute of Technology, Beijing 100081, China; 3120205167@bit.edu.cn (K.Y.); yangjiajie@bit.edu.cn (J.Y.); 2Beijing Engineering Research Center of Detection and Application for Weak Magneti Field, Department of Physics, University of Science and Technology Beijing, Beijing 100083, China

**Keywords:** MEMS gyroscope, temperature characteristics, closed-loop control, adaptive PID control

## Abstract

Microelectromechanical System (MEMS) gyroscopes are inertial sensors used to measure angular velocity. Due to their small size and low power consumption, MEMS devices are widely employed in consumer electronics and the automotive industry. MEMS gyroscopes typically use closed-loop control systems, which often use PID controllers with fixed parameters. These classical PID controllers require a trade-off between overshoot and rise time. However, temperature variations can cause changes in the gyroscope’s parameters, which in turn affect the PID controller’s performance. To address this issue, this paper proposes an adaptive PID controller that adjusts its parameters in response to temperature-induced changes in the gyroscope’s characteristics, based on the error value. A closed-loop control system using the adaptive PID was developed in Simulink and compared with a classical PID controller. The results demonstrate that the adaptive PID controller effectively tracked the changes in the gyroscope’s parameters, reducing overshoot by 96% while maintaining a similar rise time. During gyroscope startup, the adaptive PID controller achieves faster stabilization with a 0.036 s settling time, outperforming the 0.06 s of the conventional PID controller.

## 1. Introduction

Microelectromechanical System (MEMS) technology offers significant advantages such as small size, low cost, low power consumption, and high shock resistance. A MEMS gyroscope, as one of the most important devices, has wide applications in the fields of consumer electronics, automotive industry, and defense [1,2]. Especially in the defense area, there is a great demand for low bias instability, high sensitivity, and good linearity under wide temperature zone, etc. Currently, the performance of a MEMS gyroscope is restricted by device structure, control and signal conditioning circuit, and micromachining technologies [3,4,5]. The principle of MEMS gyroscopes is based on the Coriolis effect. The input angular velocity is detected by measuring the Coriolis force on a moving mass. According to the principle of the MEMS gyroscope, the larger the displacement of the drive mode, the greater the Coriolis force. To improve the signal-to-noise ratio of the detection output, typically, applying a driving force at the resonance frequency of the gyroscope’s drive mode results in a maximum displacement response. The drive mode of a tuning fork-type MEMS gyroscope can be approximated as a high-Q resonator, which is highly sensitive to the driving frequency. Even slight frequency errors can cause significant response attenuation. Therefore, the stable vibration of the mass in the drive direction is crucial for a high-performance MEMS gyroscope [6,7]. To achieve a stable vibration of the proof mass, a good control scheme is needed.

Digital circuits, due to the benefits of low noise, low temperature drift, and ease of debugging, are the dominating technology used in MEMS gyroscope control circuits of drive mode. Conventional digital closed-loop drive schemes use phase-locked loops (PLL) to track the drive frequency of the gyroscope and automatic gain control (AGC) techniques to stabilize the amplitude [8]. The control algorithms typically employ classical PID controllers because of their simple structure, good stability, and ease of adjustment. However, the device structure and materials used make them highly susceptible to temperature changes, such as Young’s modulus and thermal expansion coefficient, leading to changes in stiffness and damping, and which consequently shifts the resonance frequency. Many studies point out that temperature changes are an important reason for the decrease in gyroscope accuracy [9,10]. Since PID controller parameters are fixed after the initial setup, their capability of adapting to parameter changes caused by environmental temperature disturbances is limited.

For the purpose of improving the PID control performance, an adaptive closed-loop drive control system is proposed. The adaptive PID controller will adjust its parameters based on error values to accommodate the changing parameters of the gyroscope under temperature disturbances.

To verify the effectiveness of the designed closed-loop control system, First, simulations of temperature characteristics were conducted in COMSOL to determine key parameters of the MEMS gyroscope model. Subsequently, the simulation model of the MEMS gyroscope with temperature parameters was built in the Simulink environment of MATLAB. Finally, a closed-loop control system incorporating both PLL and AGC loops was established in Simulink, with the controller utilizing an adaptive PID algorithm.

## 2. Simulation Model of MEMS Gyroscope

### 2.1. Structure of MEMS Gyroscope

The gyroscope used in this study is a linear vibrating tuning fork gyroscope [11], whose structural model is shown in Figure 1.

The tuning fork-type MEMS gyroscope is a tuning fork type and made from monocrystalline silicon wafer bonding on glass substrate. The device structure is composed of consists of two proof movable mass blocks, with fixed anchors around them, movable beams, and comb structures [12]. The gyroscope has a completely symmetrical structure. The two proof masses vibrate in different directions and eliminate errors caused by acceleration in the detection mode direction. This allows for differential detection to eliminate errors caused by acceleration in the detection mode direction. Due to the symmetrical structure of the aforementioned MEMS gyroscope, for simplicity, the analysis can focus on one of the mass blocks, which can be equivalently modeled as a system with two mass-spring-damper models, as shown in Figure 2.

The driving force causes the gyroscope to vibrate in the X direction. When the gyroscope rotates around the Z axis, the mass experiences a Coriolis force in the Y direction. Considering coupling motion and mechanical–thermal noise, the gyroscope’s dynamic equation is given by Equation (Equation 1): (1)$$\;\left\{\begin{array}{c} m_{x}\ddot{x}+c_{xx}\dot{x}+k_{xx}x=F_{x}+2m_{c}\Omega_{z}\dot{y}\\ m_{y}\ddot{y}+c_{yy}\dot{y}+k_{yy}y=-2m_{c}\Omega_{z}\dot{x} \end{array}\right.\;$$
where $F_{d}x$ is the driving force, $\Omega_{z}$ is the angular velocity, and *x* and *y* are the displacements in the X (drive) and Y (sense) directions, respectively. $m_{x}$, $c_{x}$, and $k_{x}$ are the equivalent mass, damping coefficient, and stiffness coefficient of the drive mode. $m_{y}$, $c_{y}$, and $k_{y}$ are the equivalent mass, damping coefficient, and stiffness coefficient of the sense mode. $m_{c}$ is the equivalent mass of the Coriolis mass, approximately equal to $m_{y}$ in this model.

However, due to manufacturing errors, quadrature errors exist in the gyroscope output, primarily manifesting as coupling stiffness in the dynamic equations. Additionally, manufacturing errors can also cause coupling damping. Considering these error factors, Equation (Equation 1) is modified to Equation (Equation 2) to obtain a more accurate dynamic model.
(2)$$\;\left\{\begin{array}{c} m_{x}\ddot{x}+c_{xx}\dot{x}+c_{xy}\dot{y}+k_{xx}x+k_{xy}y=F_{x}+2m_{c}\Omega_{z}\dot{y}-F_{MNTx}\\ m_{y}\ddot{y}+c_{yy}\dot{y}+c_{yx}\dot{x}+k_{yy}y+k_{yx}x=-2m_{c}\Omega_{z}\dot{x}-F_{MNTy} \end{array}\right.\;$$
where $F_{MNTx}$ and $F_{MNTy}$ are mechanical thermal noise. $c_{xy}$ and $c_{yx}$ are damping coupling. $k_{xy}$ and $k_{yx}$ are stiffness coupling. The coupling stiffness and coupling damping can be determined by Equations (Equation 4) and (Equation 5) [13]. α and β are error angles between the actual axis and design axis.
(3)$$k_{xy}=k_{yx}=\frac{sin\;2\alpha \left(k_{xx}-k_{yy}\right)}{2}\;$$
(4)$$c_{xy}=c_{yx}=\frac{sin\;2\beta \left(c_{xx}-c_{yy}\right)}{2}\;$$

Based on Equation (Equation 2), a drive mode model of the MEMS gyroscope was established in MATLAB’s Simulink (version 9.12) environment, as shown in Figure 3.

In Figure 3, $V_{driving}$ is the model input representing the drive signal, which is converted into driving electrostatic force by a torque converter. $m_{x}$ is the drive mass. $c_{xx}$ and $k_{xx}$ are the temperature-dependent damping and stiffness coefficients of the drive mode. In Figure 3, the error terms caused by coupling stiffness and coupling damping are highlighted in green; the error terms caused by the coupling of the sensing output to the drive are highlighted in blue; and the mechanical thermal noise is highlighted in red. The impact of temperature on MEMS gyroscopes is significant. For the drive mode, both $c_{xx}$ and $k_{xx}$ in Equation (Equation 2) vary with temperature, while the equivalent mass of the drive mode, mainly composed of the Coriolis mass and the drive frame, can be considered constant. Since the coupling coefficient is relatively small, the impact of temperature disturbances on the gyroscope model is primarily due to the variations of $c_{xx}$ and $k_{xx}$ with temperature [14]. It is necessary to determine the temperature-dependent stiffness coefficient $k_{xx}$ and damping coefficient $c_{xx}$ for the drive mode and establish a MEMS gyroscope model that includes temperature parameters.

### 2.2. Relevant Parameters of MEMS Gyroscopes under Temperature Disturbances

The equivalent stiffness coefficient of the drive mode is affected by the temperature dependence of the material properties and the pre-stress changes caused by thermal expansion [15,16,17]. Material properties include Young’s modulus, density, and thermal expansion coefficient. The stiffness coefficient is typically difficult to measure directly. Equation (Equation 5) shows the relationship between the stiffness coefficient and the resonance frequency, which allows changes in the stiffness coefficient to be reflected through changes in the resonance frequency.
(5)$$\omega =2\pi f=\sqrt{\frac{k}{m}}$$

In COMSOL 6.2 software, the designed gyroscope was geometrically modeled. The model material, reference temperature, temperature coefficient of Young’s modulus, and linear expansion coefficient of the elastic material were set, and the mesh was divided. Using the finite element method, the characteristic frequencies and pre-stress studies were conducted in the solid mechanics physics field to obtain the characteristic frequencies of the gyroscope’s natural modes [18]. The equivalent mass of the drive mode was calculated based on the mechanical structure dimensions and material density. Using Equation (Equation 5), the corresponding equivalent stiffness coefficients of the drive mode at different temperatures were obtained. First, the first six characteristic frequencies of the MEMS gyroscope structure at room temperature were calculated. According to the mode shape analysis, the lowest two modes correspond to the designed drive mode and detection mode of the MEMS gyroscope, with resonance frequencies of 4565.4 Hz and 7859.2 Hz, respectively. The simulation results are ultimately consistent with the design.

By setting parameter scanning, the changes in the resonance frequency and stiffness coefficient of the drive mode with temperature variation in the range of −40 °C to 60 °C were calculated, as shown in Figure 4.

Damping in gyroscopes primarily comprises mechanical damping and air damping. Due to packaging constraints in practical engineering, even MEMS gyroscopes sealed in vacuum maintain a pressure level of approximately 10 Torr, where air damping remains predominant [18,19]. MEMS gyroscopes employ comb structures with membrane structures, thus air damping primarily manifests as membrane damping between the tops and bottoms of comb fingers. Treating the encapsulated air as an ideal gas model, the damping coefficient’s relationship with temperature is derived using Sutherland’s formula (Equation (Equation 6)) and the Clapeyron equation (Equation (Equation 7)), resulting in the expression of Equation (Equation 8).
(6)$$\frac{\eta_{p}}{\eta_{p0}}={\left(\frac{T}{T_{0}}\right)}^{1.5}\frac{T_{0}+B_{0}}{T+B_{0}}\;$$
(7)Vp=nRT
(8)$$c_{x}=\eta_{p0}{\left(\frac{T}{T_{0}}\right)}^{1.5}\frac{T_{0}+B_{0}}{T+B_{0}}\frac{nRT}{V}S_{x}\;\;$$

Here, $\eta_{p}$ and $\eta_{p0}$ are the air viscosity coefficients at temperatures *T* and $T_{0}$, respectively. $S_{x}$ is the membrane area of the drive mode comb fingers, *V* and *n* are the gas volume and moles, $B_{0}$ is the Sutherland constant related to the gas type, and *R* is the molar gas constant.

The damping coefficient of MEMS gyroscopes is primarily influenced by the membrane damping of encapsulated gases, which varies with temperature, gas tightness, and internal gyroscope structure. Consequently, precise measurement is challenging. In practical applications, the relationship between quality factor *Q* and damping coefficient $c_{x}$, as given by Equation (Equation 9), is used to reflect changes in the damping coefficient through variations in quality factor and resonance frequency:(9)$$c_{x}=\frac{\omega m_{x}}{Q}$$

To construct a damping coefficient temperature model, Equation (Equation 5) is first transformed into the form of Equation (Equation 10):(10)$$c_{x}=A_{cx}n\frac{T^{2.5}}{T+B_{0}}$$

In Equation (Equation 7), $A_{cx}$ is the damping conversion coefficient. Although temperature causes structural size changes in MEMS gyroscopes, affecting the damping conversion coefficient by less than 5‰, these structural changes can be ignored with regard to system damping. Thus, $A_{cx}$ remains unchanged with temperature variations. Since the encapsulation of MEMS gyroscopes can be considered a closed space, the quantity of gas molecules in the gyroscope remains constant. Therefore, using the quality factor of the MEMS gyroscope at room temperature as a reference and employing Equation (Equation 6), the corresponding parameters in Equation (Equation 7) are determined to solve for the damping coefficient and quality factor at various temperatures, as shown in Figure 5.

## 3. MEMS Gyroscope Closed-Loop Drive Control System

### 3.1. MEMS Gyroscope Closed-Loop Drive Control System

Due to the high quality factor of MEMS gyroscopes, a closed-loop drive system is necessary. The closed-loop drive system designed in this paper employs a method based on digital phase-locked loop (PLL) and automatic gain control (AGC) technology [20,21]. Figure 6 depicts the schematic diagram of the designed MEMS drive system, which consists of two closed-loop circuits: a frequency stabilization loop and an amplitude stabilization loop. The PLL ensures frequency tracking to maintain a resonance frequency operation of the gyroscope for optimal response, while the AGC loop adjusts the amplitude of the output electrostatic force to keep the gyroscope amplitude constant.

During the operation of the MEMS gyroscope, the signals controlled by the PLL and AGC loops are converted by a DCA into voltage signals. These signals are then applied to the MEMS gyroscope’s drive comb fingers through a push–pull drive circuit. The electrostatic force causes displacement in the drive frame and mass blocks in the drive mode direction, thereby altering the capacitance of the detection comb fingers. The MEMS gyroscope employs a membrane comb structure, exhibiting a linear relationship between capacitance and drive displacement. The capacitance signal is converted into a voltage signal by a C/V detection circuit, representing the drive response. After sampling the voltage signal with an ADC, mixing it with a reference signal generated by a DCO oscillator, and low-pass filtering, the phase and amplitude of the response signal are obtained. The fundamental principle of the PLL involves comparing the phase difference of the output signals using a phase detector. The error signal generated is filtered through a loop filter, and through negative feedback, it controls the oscillator output frequency change to achieve phase lock when the phase difference is zero. For the control system of the MEMS gyroscope, when the gyroscope is in resonance, the phase difference between the drive signal and the drive response signal is 90°. Therefore, adjusting the PLL locks the phase difference at 90° to track the gyroscope’s resonance frequency in real time. In the automatic gain control (AGC) loop, the amplitude information of the response signal is obtained through phase-sensitive demodulation. The error between the amplitude and a reference value is compared, and a PID controller is used to maintain constant vibration amplitude in the drive mode. However, classical PID controller parameters are closely related to the controlled system and do not change once set. When the temperature changes, causing gyroscope parameters to vary, the control effectiveness of a classical PID controller decreases. Traditional PID control suffers from a trade-off between overshoot and rise time. For MEMS gyroscopes, overshoot could potentially lead to comb collision damage, while prolonged rise time causes instability in the drive frequency, ultimately reducing detection accuracy. To address this, this paper proposes an adaptive PID controller design for the AGC control loop [22], capable of adapting to temperature changes. This controller suppresses overshoot while stabilizing signal amplitude as quickly as possible.

### 3.2. Adaptive PID Controller

The principle of PID control is to adjust the output signal based on the error signal. Equation (Equation 11) represents the classic PID control equation in the digital domain. In this equation, $e\left(k\right)$ is the error output at the current time step, and $u\left(k\right)$ is the control signal output at the current time step.
(11)$$u\left(k\right)=K_{p}e\left(k\right)+K_{i}\sum e\left(k\right)+K_{d}\left(e\left(k\right)-e\left(k-1\right)\right)$$

$K_{p}$ is the proportional term, where the control output is proportional to the error. As the error decreases, the control output gradually decreases and eventually stabilizes, causing the system output to converge to a stable value. The magnitude of $K_{p}$ reflects the controller’s response speed; increasing $K_{p}$ accelerates the response, but an excessively large $K_{p}$ can reduce stability and cause oscillations. $K_{i}$ is the integral term. A purely proportional term results in a steady-state error between the output and the reference value. The integral term accumulates the error over time, adjusting the control output to eliminate this steady-state error. The integral term’s main function is to eliminate the steady-state error; the larger the gain, the quicker the adjustment. However, an excessively large integral term can lead to integral saturation and cause oscillations. $K_{d}$ is the derivative term. The derivative of the error reflects the trend of the system’s response. By anticipating changes in the response, the derivative term adjusts the control signal in advance. Therefore, the derivative term can effectively reduce overshoot. However, it has the weakest ability to resist disturbances, and a large $K_{d}$ can easily lead to system instability.

In classic PID control, the proportional term plays the main control role, the integral term reduces the steady-state error, and the derivative term improves overshoot. The parameters of classic PID control are determined based on the characteristics of the controlled system, typically using the Ziegler–Nichols method in engineering. This involves first increasing the proportional gain and observing the step response to determine the critical proportional gain $K_{U}$ and recording the oscillation period $T_{U}$. Then, the appropriate PID parameters are selected according to the desired control effect. The parameters of classic PID control are closely related to the controlled system and do not change once determined. The actual control effect depends on the system parameters at the time of tuning. However, for MEMS gyroscopes, a sudden change in temperature can alter the gyroscope’s resonant frequency. Since the gyroscope operates at the resonant frequency and has a high quality factor, its response will significantly decrease, leading to errors. Under these conditions, the system parameters change, reducing the effectiveness of PID control and potentially causing instability in extreme cases. Therefore, in practical tuning, it is often necessary to sacrifice some PID control performance to achieve a more stable control system.

In the PID control process, during the initial adjustment phase, the proportional coefficient has the most significant impact. A larger proportional coefficient allows the system to reach the set value more quickly. When the system output approaches the set value, the output from the proportional term stabilizes, and the integral term becomes the primary influencing factor. The derivative coefficient functions similarly to damping, and its main effect occurs when the output is near the target. Based on this principle, the PID parameters can be set as functions of the error, with the error signal as the independent variable.

For the proportional coefficient $K_{p}$, a larger coefficient is needed when the error is large to ensure a quick response. As the system output approaches the target value and the error decreases, the response can slow down to enhance system stability. For the integral coefficient $K_{i}$, which primarily reduces steady-state error, it can be deactivated during the early adjustment phase when the error is large to prevent excessive integral output. As the error decreases, the integral action is enhanced to quickly eliminate the remaining error. For the derivative coefficient $K_{d}$, similar to damping, a smaller coefficient in the early adjustment phase avoids extending the rise time. As the error decreases and the system approaches the set value, enhancing the derivative term can significantly reduce system overshoot without compromising stability.

Based on the above analysis of the PID adjustment process, functions for $K_{p}$, $K_{i}$, and $K_{d}$ in relation to the error were constructed, enabling an adaptive PID controller that adjusts parameters according to the actual error. Equation (Equation 12) is the expression for the proportionality coefficient $K_{p}$. When the error is large, the proportionality coefficient becomes larger. When the error tends to infinity, $K_{p}$ takes the maximum value $a_{p}+b_{p}$, and when the error tends to 0, it takes the minimum value $a_{p}$.
(12)$$f_{kp}\left(e(k)\right)=a_{p}+b_{p}\left(1-e^{-c_{p}\left|e\left(k\right)\right|}\right)$$

Equation (Equation 13) is the expression for the integration coefficient $K_{i}$. When the error is small, the integration coefficient plays a major role. Therefore, at this time, $K_{i}$ takes the maximum value $a_{i}$. If the error is large, stop the integration function to avoid integration saturation.
(13)$$f_{ki}\left(e(k)\right)=\left\{\begin{array}{cc} a_{i}*e^{-c_{i}\left|e\left(k\right)\right|} & \mathrm{if}\;\left|e\left(k\right)\right|\geq \theta \\ 0 & \mathrm{if}\;\left|e\left(k\right)\right|<\theta \end{array}\right.$$

Equation (Equation 14) is the expression for the differential coefficient $K_{d}$. When the error is large, the differential coefficient is small, and as the error decreases, the differential coefficient increases. When the error approaches infinity, the differential gain takes the minimum value $a_{d}-b_{d}$, and when the error approaches zero, it takes the maximum value $a_{d}$.
(14)$$f_{kd}\left(e(k)\right)=a_{d}+b_{d}\left(1-e^{-c_{d}\left|e\left(k\right)\right|}\right)$$

### 3.3. Simulink Model of MEMS Gyroscope Closed-Loop Drive Control System

According to the schematic diagram in Figure 6, a closed-loop drive system model for MEMS gyroscope was designed in a Simulink environment, as shown in Figure 7.

In Figure 7, the gyro module is the Simulink model of the MEMS gyroscope shown in Figure 3. The damping coefficient and stiffness coefficient in the model are influenced by temperature, with specific values determined by the gyro parameter module. On the left, the phase-locked loop (PLL) circuit consists of a mixer, a low-pass filter, and a DCO. On the right, the automatic gain control (AGC) loop is composed of a mixer, a low-pass filter, and an adaptive PID module. The output of the MEMS gyroscope module is the drive displacement, which, after displacement-to-capacitance (x/C) conversion and capacitance-to-voltage (C/V) conversion, yields the drive response signal.

Table 1 shows some parameters of the Simulink model for the closed-loop drive system of MEMS gyroscope.

### 3.4. The Stability of the Drive Loop

In the gyroscope control circuit, the amplitude of the gyroscope response is controlled through an AGC circuit. According to Figure 6, the AGC control circuit can be represented as the block diagram in the following Figure 8. In the figure, H (s) is the transfer function of the gyroscope, Vac is the driving voltage, $F_{d}$ is the driving force, *x* is the mass displacement, $V_{ds}$ is the driving detection voltage, and $DV_{ds}$ is the digital driving detection voltage.

This system is a negative feedback system, with the input being a reference amplitude voltage and the output being an amplitude detection voltage. Select the corresponding open-loop analysis node at the corresponding position in the Simulink model, linearize each module, and calculate the open-loop transfer function of the system as Equation (Equation 15):(15)$$H_{OL}\left(s\right)=\frac{-5.617\times 10^{5}s^{3}-1.702\times 10^{7}s^{2}-7.131\times 10^{14}s-6.951\times 10^{15}}{s^{5}+39.81s^{4}+2.093\times 10^{9}s^{3}+4.136\times 10^{10}s^{2}+1.045\times 10^{18}s}$$

The Bode diagram of the system is shown in Figure 9, and it can be seen that there is a clear peak at the resonant frequency of the gyroscope, and the phase corresponding to the resonant point is 90°.

Figure 10 and Figure 11 show the plot-zero map and Nyquist diagram of the system.

All the poles in the figure are on the negative half of the real axis, and the Nyquist curve of the system does not enclose the reference point (−1, j0). Therefore, according to the Nyquist criterion, the closed-loop system is stable.

## 4. Results

The closed-loop drive system model of Figure 7 was simulated in a Simulink environment. Firstly, the control effect of the adaptive PID algorithm was tested when the temperature gradually increased, and the results are shown in Figure 12.

From the figure, it can be seen that the amplitude of the response signal consistently maintains the set value of 1 V without significant fluctuations, and the drive signal remains phase-locked with the detection signal at 90°. The figure also shows that to maintain a constant amplitude of the response signal, the amplitude of the drive signal gradually increases. This is because as the temperature rises, the damping coefficient increases, requiring greater driving force. Figure 13 shows the frequency of the drive signal measured using the sinusoidal analysis module. Comparing the current temperature and the gyroscope’s resonant frequency curve reveals that the drive frequency follows the changes in the resonant frequency, ensuring that the gyroscope consistently operates at its resonant frequency.

Figure 14 shows the changes in the adaptive $K_{p}$, $K_{i}$, and $K_{d}$ parameters as the temperature steadily rises. With the increase in simulation time, due to the continuous variation in error, the three PID parameters adaptively follow the changes in error, eventually stabilizing at their respective steady values as the error approaches zero.

A comparison was made between the control effects of classical PID and adaptive PID. The parameters for classical PID were determined using the Ziegler–Nichols method. Figure 15 shows the amplitude of the response signal under both control methods.

When the temperature changes steadily, the error changes at each time step are small, and both classical PID and adaptive PID can effectively follow the temperature variations. However, due to environmental temperature changes and the heating of the circuit itself, temperature fluctuations might be more pronounced. In such cases, the adaptive PID method can better accommodate parameter changes. Figure 16 shows the simulation results of the gyroscope module’s input and output when temperature jumps occur at t = 1 and t = 2. Figure 17 shows the measured response signal frequency. Figure 18 shows the three parameters $K_{p}$, $K_{i}$, and $K_{d}$.

Under conditions of temperature jumps, significant changes in error occur due to temperature variations. Comparing the classical PID controller with the adaptive PID controller, it is evident that the adaptive PID controller better adapts to parameter changes, maintaining stable driving amplitude. The control effects comparison is shown in Figure 19.

The quality factor of a gyroscope is mainly determined by damping, so in simulation, reducing the damping coefficient can improve the quality factor of the gyroscope. Figure 20 shows a comparative simulation between the adaptive PID controller and the classical PID controller when the quality factor is around 10,500. Although the overshoot of the system has slightly increased due to reduced damping, it still has better control performance compared to classical PID.

Comparing the control effects of classical PID and adaptive PID, from the gyro initiation to the steady state at room temperature, the classical PID has a rise time of approximately 0.006 s but an overshoot of 0.6 V, with a settling time of 0.06 s. In contrast, the adaptive PID has a rise time of 0.016 s and a settling time of 0.036 s, but the overshoot is reduced to 0.025 V, a reduction of about 96%.

## 5. Conclusions

This paper analyzes the influence of temperature on the parameters of MEMS gyroscopes through their dynamic equations. The changes in stiffness and damping coefficients with temperature were determined using finite element simulation. Based on this, a driving control system model for MEMS gyroscope was designed in a Simulink environment, and the influence of temperature changes on the control system was considered. To address the issue of poor control performance of the classical PID method due to variable model parameters, an adaptive PID controller was designed to adjust PID parameters based on error factors. Simulation experiments demonstrate that the designed closed-loop drive system possesses good control capability, enabling the gyroscope vibration to be controlled with smaller overshoot and faster stabilization speed. Compared to classical PID control, the overshoot is reduced by approximately 96%, while maintaining a similar rise time.

## Figures and Tables

**Figure 1 micromachines-15-01102-f001:**
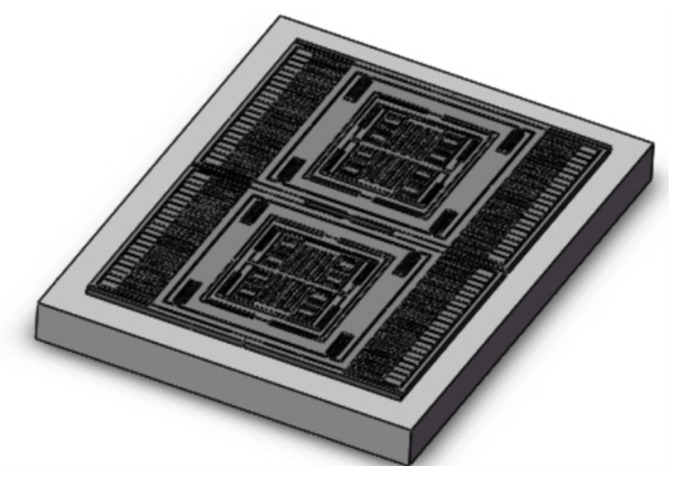
Structural model of tuning fork gyroscope.

**Figure 2 micromachines-15-01102-f002:**
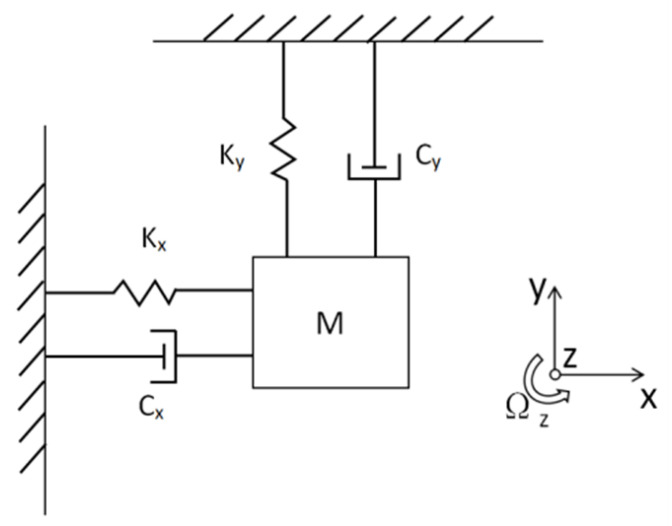
Simplified gyroscope dynamic model.

**Figure 3 micromachines-15-01102-f003:**
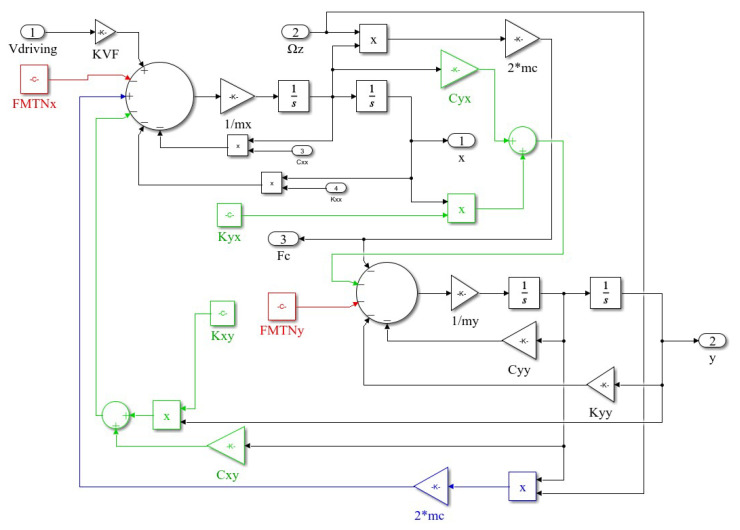
Simulink model of MEMS gyroscope driving mode.

**Figure 4 micromachines-15-01102-f004:**
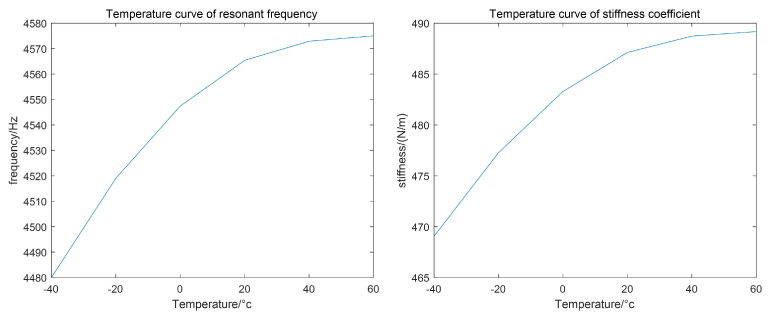
The variation curve of resonant frequency and stiffness coefficient caused by temperature.

**Figure 5 micromachines-15-01102-f005:**
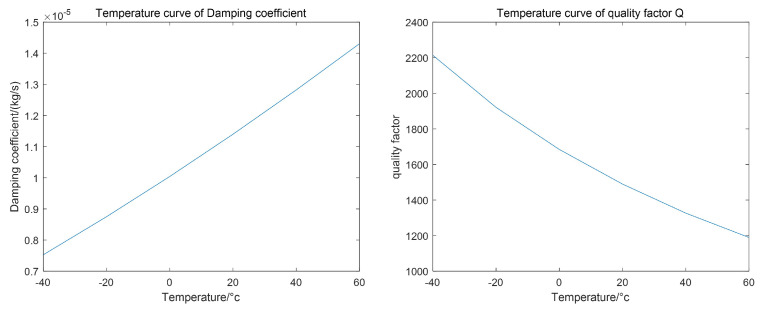
The variation curve of damping coefficient and quality factor caused by temperature.

**Figure 6 micromachines-15-01102-f006:**
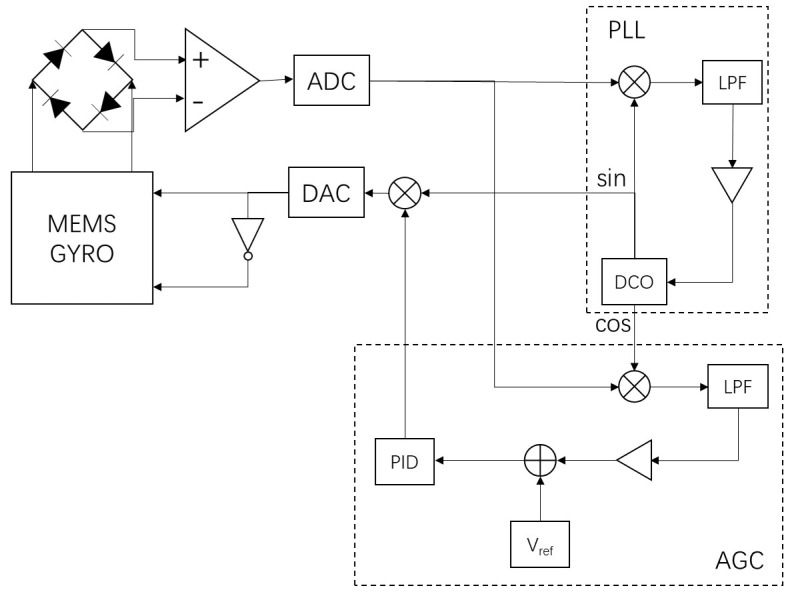
Schematic diagram of MEMS gyroscope closed-loop drive system.

**Figure 7 micromachines-15-01102-f007:**
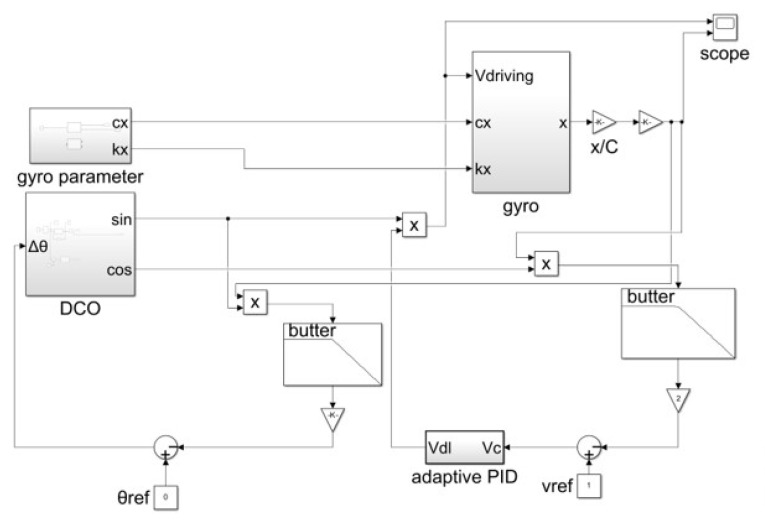
Simulink model of MEMS gyroscope closed-loop drive system.

**Figure 8 micromachines-15-01102-f008:**
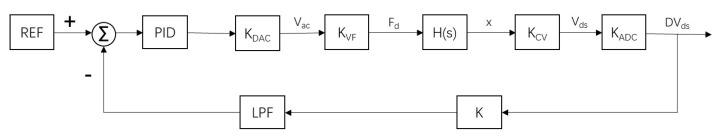
The AGC control circuit.

**Figure 9 micromachines-15-01102-f009:**
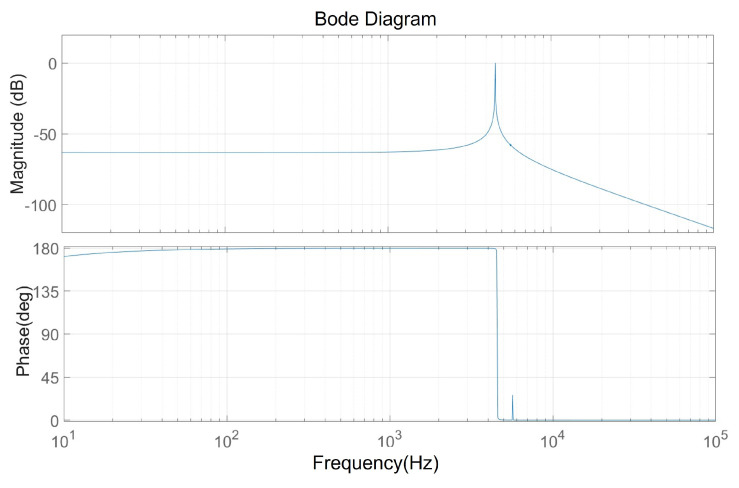
Bode diagram.

**Figure 10 micromachines-15-01102-f010:**
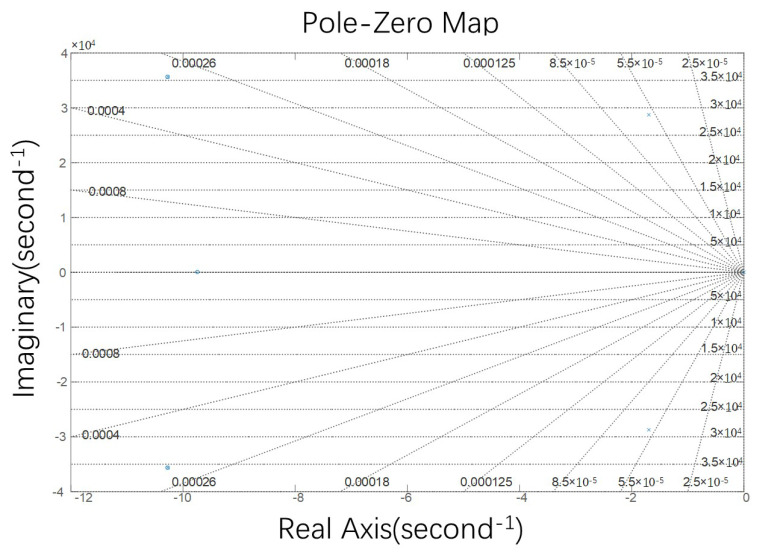
The plot-zero map.

**Figure 11 micromachines-15-01102-f011:**
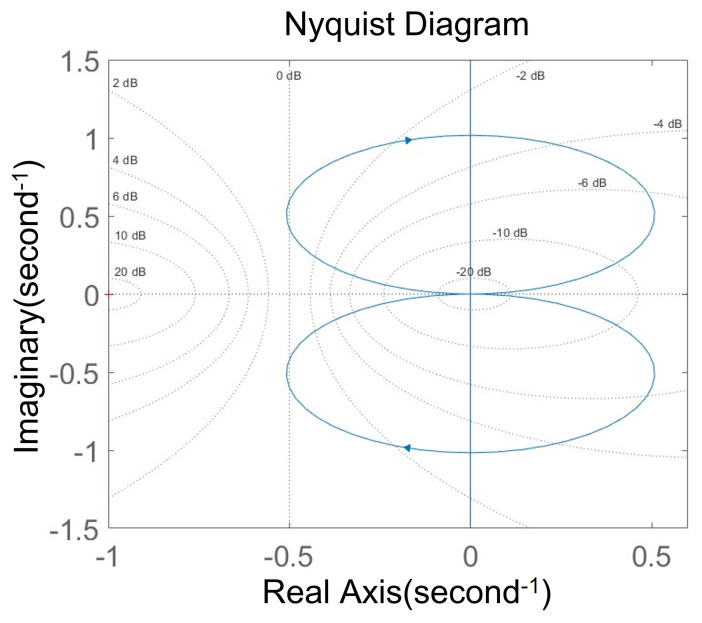
Nyquist digarm.

**Figure 12 micromachines-15-01102-f012:**
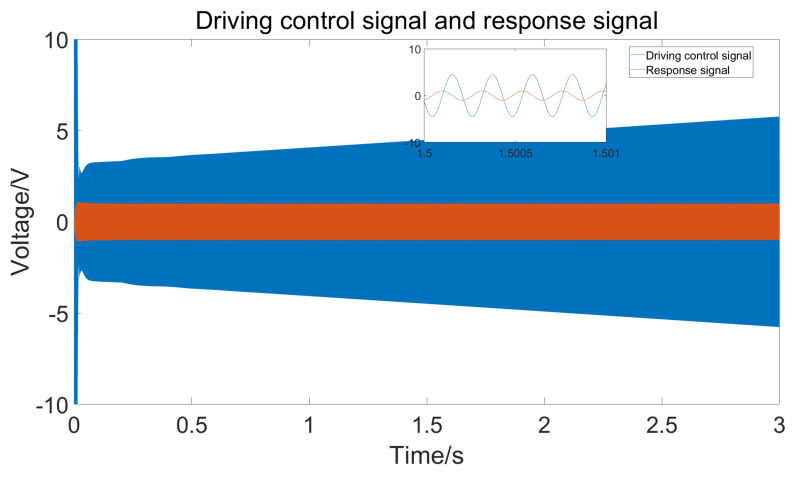
Driving control signal and response signal under stable temperature changes.

**Figure 13 micromachines-15-01102-f013:**
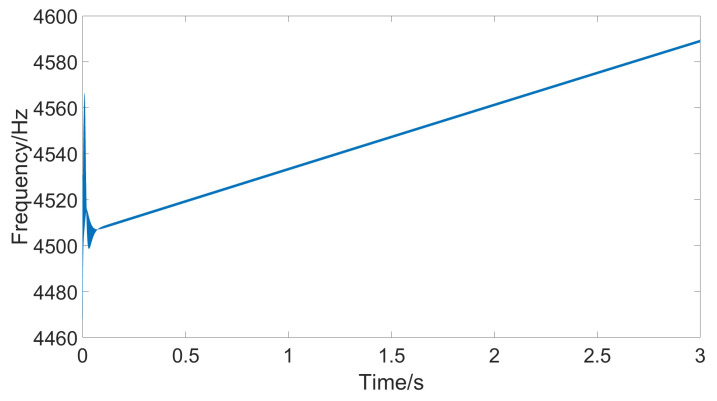
Resonant frequency under stable temperature changes.

**Figure 14 micromachines-15-01102-f014:**
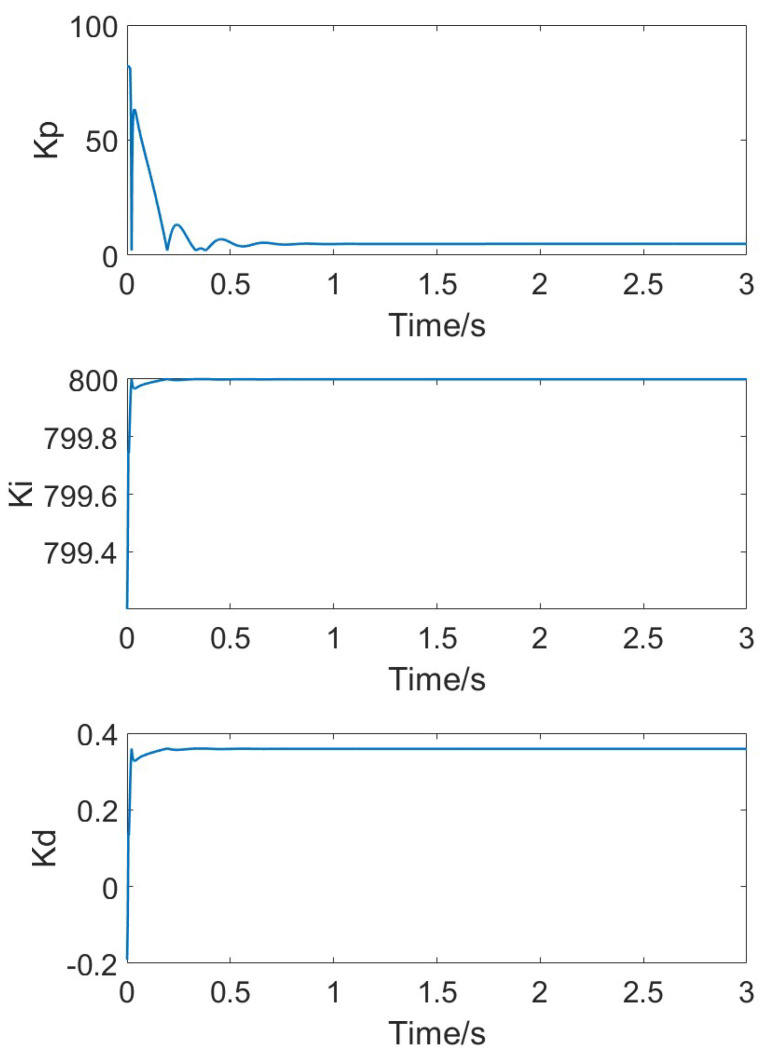
Adaptive parameters of $K_{p}$, $K_{i}$, and $K_{d}$ under stable temperature changes.

**Figure 15 micromachines-15-01102-f015:**
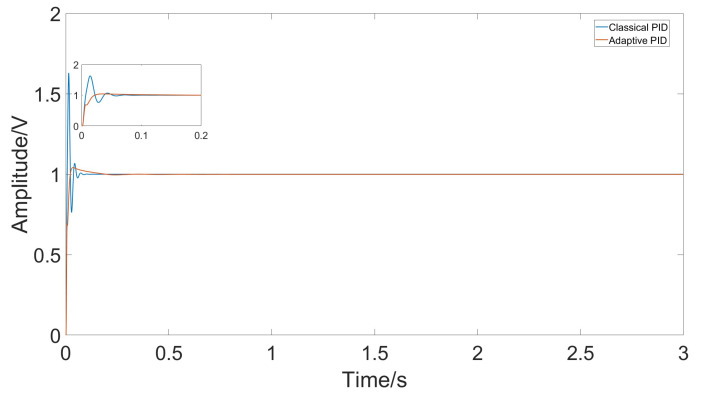
Response amplitude of classical PID and adaptive PID under stable temperature changes.

**Figure 16 micromachines-15-01102-f016:**
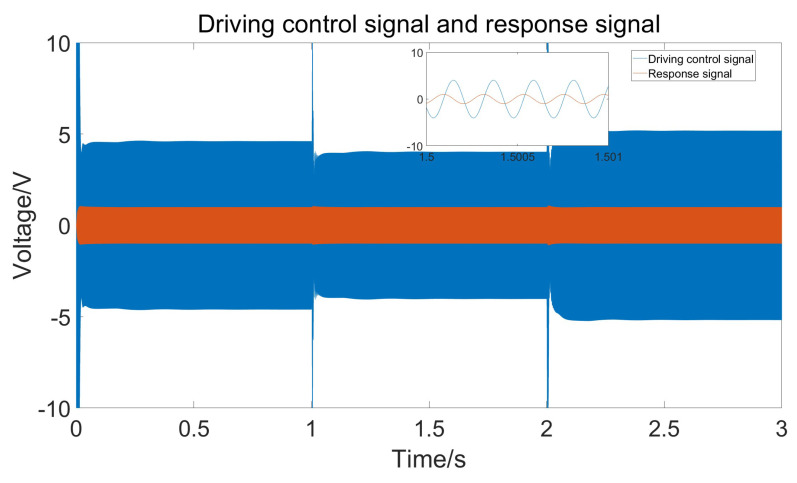
Driving control signal and response signal under rapid temperature changes.

**Figure 17 micromachines-15-01102-f017:**
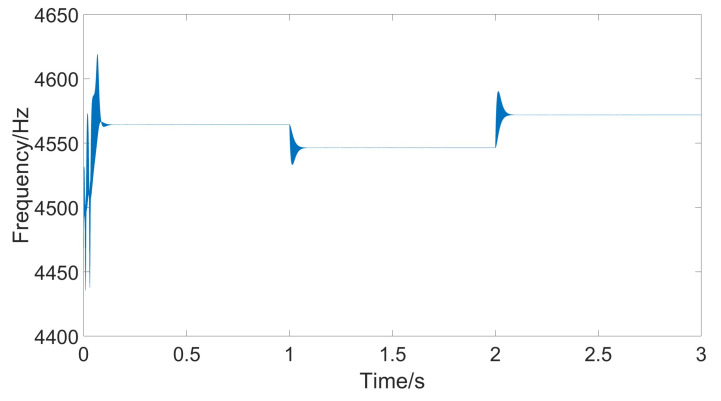
Resonant frequency under rapid temperature changes.

**Figure 18 micromachines-15-01102-f018:**
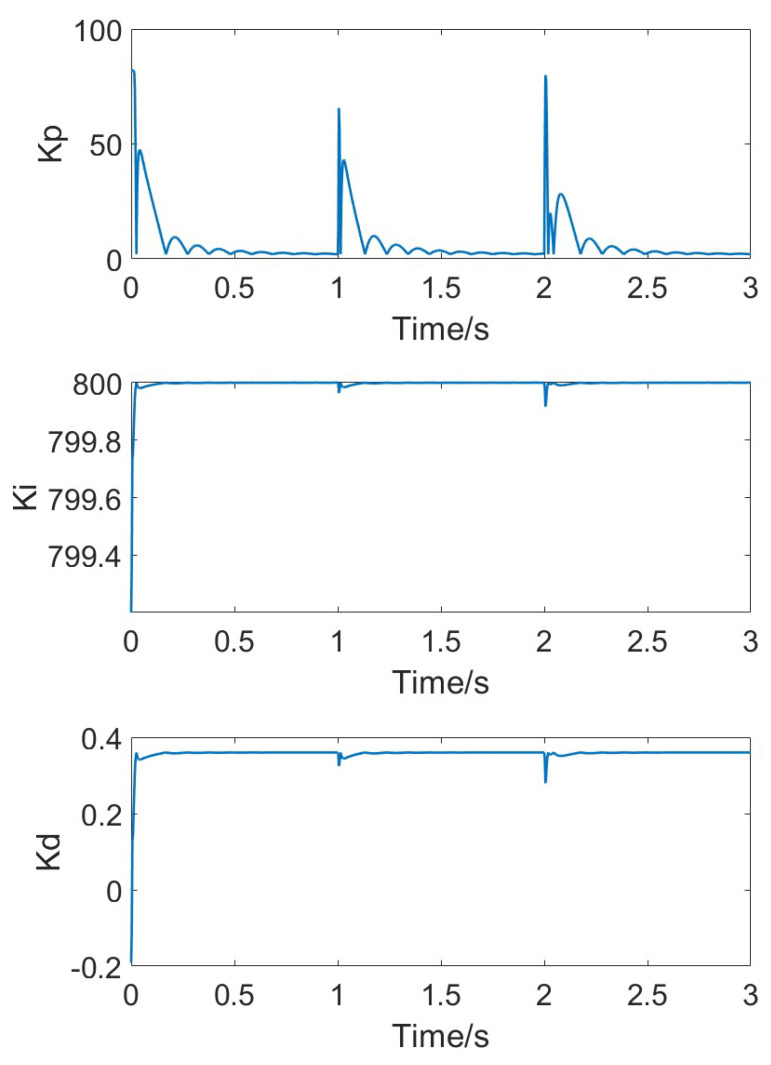
Adaptive parameters of $K_{p}$, $K_{i}$, and $K_{d}$ under rapid temperature changes.

**Figure 19 micromachines-15-01102-f019:**
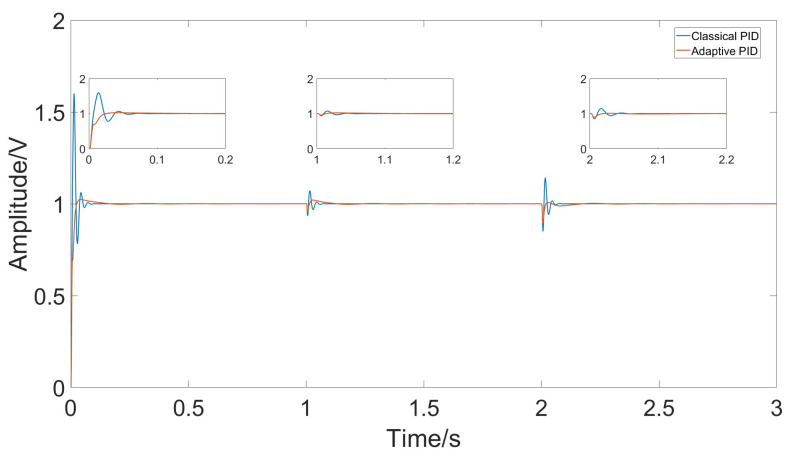
Response amplitude of classical PID and adaptive PID under rapid temperature changes.

**Figure 20 micromachines-15-01102-f020:**
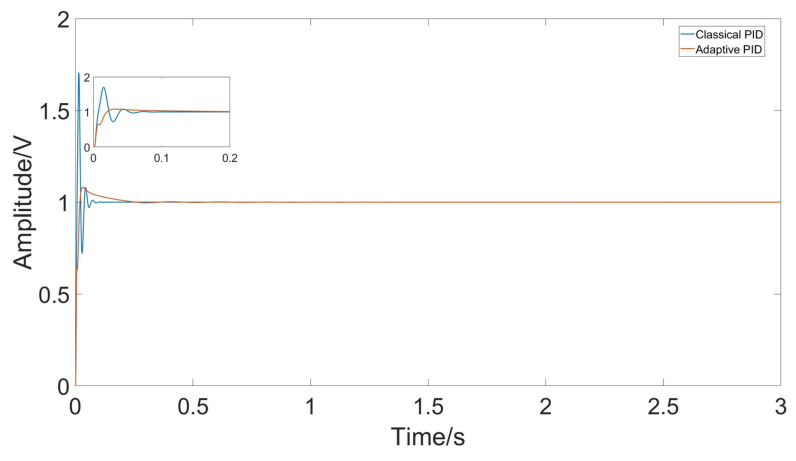
Response amplitude of classical PID and adaptive PID under rapid temperature changes.

**Table 1 micromachines-15-01102-t001:** Simulink model parameters of the MEMS gyroscope closed-loop drive system.

Parameters	Unit	Value or Expression
$m_{x}$	kg	$5.92\times 10^{-7}$
$c_{xx}$	N/m/s	$1.013\times 10^{-5}+6.781\times 10^{-8}$ T
$k_{xx}$	N/m	480.5+0.1982 T
$c_{xy}$/$c_{yx}$	N/m/s	$3.49\times 10^{-8}$
$k_{xy}$/$k_{yx}$	N/m	6.98
$F_{MNTx}$	N	$8.89\times 10^{-13}$
$F_{MNTy}$	N	$9.05\times 10^{-13}$
$K_{VF}$	N/V	$10^{-8}$
$K_{xC}$	F/m	$8.5\times 10^{-7}$
$K_{CV}$	F/V	$-8\times 10^{12}$

## Data Availability

Data are contained within the article.

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
