# Peer review of "Research on Adaptive Closed-Loop Control of Microelectromechanical System Gyroscopes under Temperature Disturbance"

_micromachines, 2024, doi:10.3390/mi15091102_

Round 1

Reviewer 1 Report

Comments and Suggestions for Authors

1. "Microelectromechanical Systems (MEMS) gyroscopes are essential for angular rate detection in inertial navigation.“ essential?It is well known that there is far more than one type of gyroscope for measuring rates in inertial navigation systems, and that other gyroscopes typically outperform MEMS gyroscopes.

2. Please rewrite the abstract to enhance readability.

3. The introduction has only one paragraph, please describe  background, others‘ research and the work of this paper in separate paragraphs.

4. The comsol simulation presented in the abstract is not prominent in the paper.

5. What is ”DCA“ in figure 6?

6. As shown in Figure 5, the quality factor is 2000, which is too low for the vast majority of current MEMS gyroscopes.

7. The work in this paper is simulation only, please include more non-ideal factors such as temperature drift in gain. And the dynamic equations are too ideal.

8. Please includean illustration for the stability of the drive loop.

Comments on the Quality of English Language

Please revise the professional vocabulary. detection mode? rate detection?

Author Response

Thank you very much for taking the time to review this manuscript. Your suggestions have greatly contributed to improving the quality of this paper. In particular, you have meticulously pointed out some overlooked issues regarding the professionalism of the language and the logic of the expression. I have made the necessary revisions to the paper based on the issues you highlighted in your comments. See Attachment    

Reviewer 2 Report

Comments and Suggestions for Authors

This paper utilizes an adaptive PID control method to improve its stabilization speed and eliminate the overshoot.  Please find below a few suggestions that can help to improve the quality of the paper:

1. The gyroscope itself is reused from a previous paper which has been published in 2019, so its design and modeling details are not new to this paper. If those details have to be included here, proper references are needed.

2. Is the gyro fabricated? Please include the measurement data if possible. Also, the FEA & Simulink modeling results need to be validated against the measurement data.

3. Need to unify the format of all the plots to make them more readable (font size, line width...)

Comments on the Quality of English Language

The English is good and readable

Author Response

Thank you very much for taking the time to review this manuscript. Your suggestion is very helpful in improving the quality of my paper.

I have answered each of the questions you mentioned below and marked which parts of the paper have been modified. Thanks.

Round 2

Reviewer 1 Report

Comments and Suggestions for Authors

1. Why don't use PI controllers, please explain why the differential link is added.

2. What is F_MNT in equation (1)?

3. The non-ideal dynamical equations should contain at least damping coupling and stiffness coupling.

4. It is not enough to compare the overshoot in the abstract.

5. Why not use a higher quality factor for the simulation? The results would be more generally applicable to most MEMS gyros.

Comments on the Quality of English Language

no comments

Author Response

Thank you very much for taking the time to review this manuscript. Your suggestions have greatly contributed to improving the quality of this paper. I have made the necessary revisions to the paper based on the issues you highlighted in your comments. As follows.

  1. Why don't use PI controllers, please explain why the differential link is added.

In PID control, the differential control has the function of leading adjustment. The differential term reflects the rate of change in the system's deviation signal, allowing for prediction of the trend of deviation changes and thus acting in advance. This is equivalent to increasing the damping of the system, thereby reducing overshoot during adjustment and stabilizing fluctuations in steady state conditions. The disadvantage of differential control is that high-frequency noise can cause large jumps in the rate of deviation signals, affecting the output of the controller. Therefore, careful selection of differentiation parameters is required. The parameters of the PID controller in this article are determined using the Ziegler–Nichols method.

  1. What is F_MNT in equation (1)?

Thank you for pointing out. I forgot to mark it in the manuscript. Here, F_MNT is a quantified mechanical thermal noise force.

  1. The non-ideal dynamical equations should contain at least damping coupling and stiffness coupling.

Thank you for your suggestion. Although the gyroscope uses a cantilever beam structure to achieve decoupling between two modes, due to manufacturing errors and other factors, coupling stiffness and damping are inevitable. Especially the coupling stiffness, which is the manifestation of quadrature error in the mechanical model of gyroscope. Therefore, I have referred to the analysis of quadrature errors in reference [13] and added expressions for coupling stiffness and damping in this paper.

4.It is not enough to compare the overshoot in the abstract.

Thank you for your suggestion. I have revised the abstract and conclusion sections, and added the parameter 'settling time' to analyze the control performance. Settling time refers to the time required for the system output to adjust from a deviated state to a stable state. Using an adaptive PID controller can significantly reduce the settling time.

  1. Why not use a higher quality factor for the simulation? The results would be more generally applicable to most MEMS gyros.

Thank you for your suggestion. In the dynamic model of the gyroscope, the quality factor is primarily determined by the damping coefficient. Therefore, in the simulation, I reduced the damping coefficient and simulated the effect of using an adaptive PID controller with a quality factor of approximately 10500. Although the system's overshoot slightly increased due to the reduced damping, the adaptive PID control still outperforms the conventional PID in terms of control effectiveness.

Reviewer 2 Report

Comments and Suggestions for Authors

Comments addressed 

Author Response

Thank you for taking the time to read my manuscript. I mainly made the following modifications in this round: 

1.Revised the abstract again and added the comparison results of settling time.

2.The simulation model of the gyroscope has been modified in the Method section to make it closer to the real situation.

3.Modified the closed-loop stability analysis. 

4.The image format has been unified, making the text in the image clearer.